Evaluation of dentin features in teeth after caries removal by three techniques (chemomechanical, mechanical with a smart bur, and air-abrasion): an in vitro study

Abdul-Kareem Mahmood Maha 1
Khairi Al-Hadithi Haraa 1
Mueen Hussein Hashim hashimmueenhussein@gmail.com hashimmueenhussein@uomustansiriyah.edu.iq 2
1 Department of Orthodontics, Pedodontics and Preventive Dentistry, College of Dentistry, Mustansiriyah University , Baghdad , Iraq
2 Department of Conservative Dentistry, College of Dentistry, Mustansiriyah University , Baghdad , Iraq
Kanavakis Georgios
Electronic publication date: 2024 Jul 15
Publication date: 2024
Volume: 12
Electronic Location ID: e17717
Received 2024 Jan 15; Accepted 2024 Jun 18
Copyright: ©2024 Abdul-Kareem Mahmood et al.
Copyright year: 2024
Copyright holder: Abdul-Kareem Mahmood et al.
License: This is an open access article distributed under the terms of the Creative Commons Attribution License, which permits unrestricted use, distribution, reproduction and adaptation in any medium and for any purpose provided that it is properly attributed. For attribution, the original author(s), title, publication source (PeerJ) and either DOI or URL of the article must be cited.
License URL: https://creativecommons.org/licenses/by/4.0/

Keywords: Dental caries, Chemomechanical caries removal, Scanning electron microscope, Air abrasion

Funding: The authors received no funding for this work.

==============================
Background

Different methods for removing dental carious lesions exist, including conventional rotary caries removal and new advanced technology like polymer-based burs, chemomechanical agents, air abrasion, and laser.

Objectives

This study shows the differences in features of dentin (smear layer, patency of dentinal tubules, surface irregularities, intertubular micro porosities, and exposed dentinal tubules) among different types of caries removal techniques.

Materials and Methods

An in vitro study was done on 60 primary molars with occlusal class I active caries. Teeth were divided into three groups according to a method of caries removal (G1: chemomechanical, G2: mechanical with a smart bur, and G3: air-abrasion). After complete caries excavation, the teeth were examined under a scanning electronic microscope (SEM) with the power of magnification 4,000x and 8,000x to show the morphological dentinal features with SEM microphotographs. Data obtained was analyzed using the SPSS program where Fisher exact, Kruskal–Wallis and multiple Wilcoxon sum rank tests were used. The level of significance is when the p-value is less than 0.05.

Results

Generally, SEM showed the highest ratio of score 1 of smear layer presence, surface irregularities, and microporosity in all groups in both magnifications. The patency of tubules showed the highest ratio of score 1 in G1, scores 2 in both G2 and G3 in magnification 4,000x, while 8,000x there was the highest ratio of its score 1 in G1 and G2 while the G3 has score 2 as the highest score. The exposed dentinal tubules showed the highest ratio in G1 in score 3, in G2 in score 2, and in G3 in score 1 in magnification 4,000x, while 8,000x there was the highest ratio of its score 2 in both G1 and G3 while the G3 has highest score 1. The study with magnification 4,000x showed a significant difference (S) among three groups in exposed dentinal tubules with a p-value (0.012), and there was S between chemomechanical and smart, chemomechanical and air-abrasions groups with a p-value (0.041, 0.001 subsequentially). Other dentin features showed non-significant differences (NS) among or between groups in both magnifications (4,000x, 8,000x).

Conclusions

All groups were effective in removing caries and can successfully treat young, scared or stressed patients. All methods of caries removal produce clinically parametric changes in the residual dentin.

Introduction

Dental caries, an irreversible disease that originates from microbes, affects the mineralized tissues of teeth. Cavity development is a typical result, and demineralization and organic tooth disintegration are its defining features (Nomura et al., 2020; Hussein, Mahmood & Alberaqdar, 2015). In most cases, drilling removes caries (Elkholany et al., 2009). However, a drill can effectively remove harmful patient experiences (Arora et al., 2012). Consequently, it is crucial to address anxious and uncooperative youngsters using a painless, noninvasive method (Kotb et al., 2009; Saeed, Hussein & Mahmood, 2017; Hussein, Saeed & Al-Zaka, 2017).

The gold standard for treating cavities was the mechanical method. However, this approach has a few drawbacks, such as not distinguishing between good and diseased tooth tissue, which removes both. Additionally, children and their moms experience psychological anguish during caries removal using the rotary method (Abdelaziz, Badran & Allam, 2022; Shah et al., 2020).

When dealing with anxious or resistant children, it is crucial to keep their comfort and the need for minimal involvement in mind (Shah et al., 2020). One alternative to traditional methods of treating active caries is chemomechanical caries removal (CMCR). The only dental tissues removed in this process are the diseased ones. Therefore, patients experience less discomfort and less anxiety due to the reduction of handpiece vibration, the avoidance of pulpal tissue irritation, and the removal of healthy dental tissue (Abdelaziz, Badran & Allam, 2022; Mahadevan & Parikh, 2010; Shah et al., 2020). The CMCR method dissolves the carious tissue. Sodium hypochlorite-based treatments (such as GK-101E, Carisolv, and Cariemove) and enzyme-based agents (such as Papacarie, Carie-care, Biosolv TM, and Brix 3000) were the two main categories of CMCR agents (Abdelaziz, Badran & Allam, 2022; Hamama, Yiu & Burrow, 2014).

Polymer (ceramic) burs can excavate carious dentin instead of the more common stainless-steel burs (Schwendicke et al., 2018). Due to the material’s hardness, which does not permit excessive preparation, their primary benefit is the selective removal of only the affected dentin. After the soft, diseased dentin is out, the instruments will become dull on the firm, healthy dentin (Marques et al., 2020). Although polymer burs take twice as long to eliminate caries as diamond or steel burs, they are more selective in the tissues they remove from the teeth (Banerjee, Kidd & Watson, 2000a; Banerjee, Watson & Kidd, 2000b).

Prevention, remineralization, and minimally intrusive therapy are the tenets of minimal intervention dentistry. It does the job with minor damage to healthy tooth tissue and the least invasiveness. It incorporates novel approaches such as air abrasion (Showkat et al., 2020). One way to remove hard tissue from teeth is by an air abrasion, a non-rotary, pseudo-mechanical technique (Banerjee, Watson & Kidd, 2000b). It uses a thin stream of compressed air containing aluminium oxide particles to destroy tooth structure. This technique protects healthy, mineralizable tissue while removing cavities selectively until the tooth surface is covered in hard, dry dentin. How well air-abrasion devices work depends on the material’s hardness and the settings used to operate them (Tan, Hill & Anderson, 2015). Particle stream precision, low air pressure (1.5–3 bars), and sufficient suction are all provided by specialized nozzles. Dentin caries lesions in occlusal pits and fissures and those on flat surfaces can be treated with air abrasion. Adhesive materials are the sole option for repair following air-abrasion caries eradication. A rubber dam and specialized face masks are necessary to reduce the likelihood of the patient’s inhaling particles. There is no need to worry about overheating, vibration, or mechanical stimuli with air-abrasion, and the patient’s behavior is more cooperative overall (De Almeida Neves et al., 2011).

After carious excavation, the adhesive system employed and the residual dentin substrate both play a role in determining the degree of adhesion between dentin and restorative material. Therefore, the purpose of this in vitro study was to evaluate and compare the residual dentin surface (smear layer, dentinal tubule patency, surface irregularities, intertubular micro porosities, and exposed dentinal tubules) after caries excavation with chemomechanical, mechanical with a smart bur, and air-abrasion using scanning electron microscope (SEM) in primary teeth. The null hypothesis was that there were no differences among the three methods of caries removal.

Materials and Methods

Study type

The experimental study was carried out on extracted primary teeth with active caries occlusal taken from children aged 6–10 years after extraction.

Ethical approval

The study was done after approval from the Committee of the College of Dentistry in the University with REC reference REC135, study number MUPEDO3, on 8-3-2023; each tooth was taken after receiving written informed consent from the parents of the children to use their teeth for dental research purposes.

Collection of teeth

Immediately after the extraction, teeth were cleaned with a Comfort-Sonic P6 Plus ultrasonic scaler (Beyes, Morrisburg, ON, Canada) to remove any remaining soft tissue debris and examined for caries detection. The caries was diagnosed using the detection mode of Eighteeth Curing Pen (Eighteeth, Changzhou City, China) with light intensity 600 mW/cm2, wavelength: 385 nm–410 nm., then the teeth were stored in a container filled with deionized water at four °C for a maximum of 7 days before carious treatment. The time of collection of teeth required in the study was about four months between 9-3-2023 and 10-7-2023.

Inclusion criteria

Extracted primary molars were without external or internal resorption of roots and with active occlusal caries (class I) extended in dentin approximately 1–2 mm below the central fissure that was assessed with a dental explorer (Hu-Friedy, Chicago, IL, USA) and spread with a distance 1 mm away from pulp are that confirmed by digital radiography X-ray with NanoPix Digital Sensor (Eighteeth).

Exclusion criteria

Teeth with deep dentinal caries approaching or involving pulp, teeth with external or internal resorption of roots, and medically compromised children’s teeth.

Sample size

It was determined using G power 3.0.10 (Program written by Franz-Faul, University of Kiel, Germany) with the power of study = 85%, alpha error of probability = 0.05 two-sided, effect size of Cohen W is 0.5 (Large effect size), where Cohen W (small = 0.1, medium = 0.3, large = 0.5) (Cohen, 1988; Faul et al., 2009). Thus, the sample size was 60 (20 samples for each).

Selection of technique of caries removal

In this study, three techniques or methods for caries removal were chosen based on how the caries was removed, where the first method includes mixed chemomechanical action (chemical material: Brix 3000 gel and mechanical blunt spoon excavator), and the second method was pure new mechanical (ceramic smart bur), and third one with more minimal intervention dentistry novel method (air abrasion that used aluminium oxide powder to remove the caries). In addition to the above more literature talks about the efficiency and advantages of these methods in the removal of caries (Banerjee, Kidd & Watson, 2000a; Banerjee, Watson & Kidd, 2000b; De Almeida Neves et al., 2011; Hamama, Yiu & Burrow, 2014; Muliyar et al., 2014; Tan, Hill & Anderson, 2015; Schwendicke et al., 2018; Marques et al., 2020; Showkat et al., 2020; Abdelaziz, Badran & Allam, 2022; Dhanvijay et al., 2023).

Sample grouping

The total 60 teeth were divided randomly allotted into three groups (20 teeth for each group) according to a technique of caries removal; G1: Chemomechanical caries removal (CMCR), used at first caries solvent is Brix 3000 (Papain 30000 U/mg 10%), then spoon excavator, G2: Mechanical caries removal (MCR): Smart ceramic polymer bur, G3: Air-abrasion caries removal (AACR).

Method of excavation of caries

The excavation of the cavities was carried out following the instructions provided by the manufacturer for each procedure. In all three groups, after complete caries removal, the assessment of healthy dentin without caries was performed visually with the naked eye, tactilely with a dental explorer, and with a Curing Pen device.

In group 1 (CMCR), the Brix 3000 gel (Brix SRL, Cipolletti, Argentina) (Fig. 1) was allowed to sit undisturbed for 30 to 60 s on top of the tooth’s carious lesion. A blunt conventional spoon excavator (Medesy srl, Maniago, Italy) was used to extract carious dentin, and this caused the dentin to soften. The caries excavation was halted at this point since the applied CMCR gel was no longer cloudy; this procedure was repeated twice or thrice. At this point, the carious dentin was effectively removed. After removing the gel residue, the cavity was quickly cleaned with water and a sterile, damp cotton pellet was used to wipe it down. The CMCR agent’s non-turbid appearance and consistent colour served as the basis for visually evaluating full caries excavation.

Figure 1 Brix 3000 gel.

In the second group (MCR), caries removal was done using a low-speed handpiece (NSK, Japan) with a ceramic round bur (Verdent, Ludowa, Poland) (Fig. 2).

Figure 2 Smart ceramic polymer bur.

In the third group (AACR), AquaCare air-abrasion (Velopex, London, UK) was used according to manufacturers using air flow prep K1 Max, EMS at 5 bar, powder flow rate 3 g/minute, water flow rate 100 mL/minute, average grain size 36 µm, 2–3 mm distance, angle 45°–70° (Fig. 3).

Figure 3 Schematic diagram of the mechanism of air abrasion.

Method of examination of dentinal features

Once all cavities had been excavated, the tooth samples were placed in distilled water and put through an ultrasonic bath BTX600 (Zhejiang, China) for five minutes. Later, the specimens were overnight dried in a desiccator (Shanghai, China) until the water residue was removed. Then, Axia™ ChemiSEM™ Scanning Electron Microscope (SEM) (Thermo Fisher Scientific) with the power of magnification (4,000x, 8,000x) was used to evaluate the clinical characteristics of dentinal features (Thakur et al., 2017). Then, the tooth samples were placed into a scanning electron microscope (SEM) vacuum chamber and coated with a gold sputtering unit. The scanning electron microscope revealed the cavity’s innermost depths. Microphotographs were taken at 4,000x and 8,000x magnifications to view the surface morphology and five distinct locations were examined. All the microphotographs included in the analysis were shot at the working distance (WD = 17–19.8 mm), and the scale bar was presented in all SEM snaps as Axia ChemiSEM (Thermo Fisher Scientific) which was 20 µm in Fig. 4 (SEM microphotograph of a sample under 4,000x magnification) and 50 µm in Figs. 4–6 (SEM microphotograph of a sample under 8,000x magnification). SEM ultra-morphologic analyses were conducted using the remaining tooth structure (smear layer, patency of dentinal tubules, surface irregularities, intertubular micro-porosities, and exposed dentinal tubules) (Thakur et al., 2017; Thazhatheethil et al., 2021). The presence of a smear layer was graded into four scores (score 0: absence of smear layer, score 1: moderate smear layer, score 2: dense smear layer with visible dentinal tubules, score 3: thick smear layer with no visible dentinal tubules). Patency of dentinal tubules scores are: score 0: absence of patent dentinal tubules, score 1: minimal number of patent dentinal tubules, score 2: moderate number of patent dentinal tubules, score 3: All the dentinal tubules were patent. Surface irregularity scores are: score 0: smooth surface, score 1: partially irregular surface, score 2: thoroughly roughened surface, score 3: roughened surface with globular projections. Intertubular micro-porosities scores: score 0: absence of intertubular micro-porosities, score 1: minimal micro-porosities, score 2: moderate micro-porosities, score 3: abundant micro-porosities. Exposed dentinal tubules scores: score 0: absence of dentinal tubules, score 1: minimal number of exposed dentinal tubules, score 2: moderate number of exposed dentinal tubules, score 3: all the dentinal tubules were exposed (Thazhatheethil et al., 2021; Somani et al., 2015). Two blinded investigators (interexaminer) evaluated and compared the SEM microphotographs.

Figure 4 SEM microphotograph of a sample under 4,000x magnification showing a dense smear layer with visible dentinal tubules.

Figure 5 SEM microphotograph of a sample under 8,000x magnification showing a dense smear layer with visible dentinal tubules.

Figure 6 SEM microphotograph of a sample under 8,000x magnification showing patent dentinal tubules with collagen network.

Data analysis

The data collected were entered in Office Excel 2019 (Microsoft, US) and then analyzed using Statistical Package for Social Science (SPSS) version 22 software (Chicago, IL, USA). Frequency, percentage, minimum, maximum and mean rank were obtained.

The data of this study after analysis was non-parametric so, Kruskal–Wallis tests were used for multiple comparisons of mean rank values of clinical features of residual dentin. Fisher’s exact test is used to assess the independent comparison among clinical features of residual dentin features or scores variables (smear layer, patency of dentinal tubules, surface irregularities, Intertubular micro porosities, and Exposed dentinal tubules) in a comparison study among three groups that are not correlated.

Fisher exact and Kruskal–Wallis tests were used; Fisher’s exact test assesses the null hypothesis of independence by applying hypergeometric distribution of the numbers in the scores of the tables. Multiple Wilcoxon sum rank tests were used and adjusted by the Dunn–Bonferroni test. The level of significance is when the p-value is less than 0.05; when the p-value of the degree of significance was equal to or more than 0.05; the significance was non-significant statistically but when the level of p-value was less than 0.05 this meant the degree of difference was significant.

Results

Clinical features of dentin with 4,000x magnification in all groups

Our results showed the ratio of score 1 of the smear layer (moderate smear layer) was higher than other scores (2,0,3) sequentially in all groups, where score 1 was 55% in G1, 50% in G3, 45% in G2. The lowest score of the smear layer was score 3 (dense without visible tubules), which was 5% in G2, and 0% in both G1 and G3 (Fig. 4, Table 1).

Table 1 Comparison of clinical features of residual dentin at 4,000x magnification among groups using Fisher exact test.

Variance	Scores	G1	G2	G3	Fisher exact	P-value	
		N	%	N	%	N	%			
Smear layer	0	4	20	4	20	5	25	2.462	0.989
NS	
1	11	55	9	45	10	50	
2	5	25	6	30	5	25	
3	0	0	1	5	0	0	
Patency of Dentinal tubules	0	3	15	3	15	3	15	2.805	0.963
NS	
1	9	45	7	35	8	40	
2	7	35	10	50	9	45	
3	1	5	0	0	0	0	
Surface Irregularities	0	6	30	5	25	4	20	4.075	0.757
NS	
1	12	60	14	70	16	80	
2	1	5	1	5	0	0	
3	1	5	0	0	0	0	
Intertubular micro porosities	0	4	20	1	5	0	0	7.308	0.217
NS	
1	8	40	10	50	13	65	
2	8	40	8	40	7	35	
3	0	0	1	5	0	0	
Exposed dentinal tubules	0	0	0	1	5	1	5	14.353	0.012
S	
1	3	15	4	20	9	45	
2	4	20	10	50	7	35	
3	13	65	5	25	3	15	
Notes.

N number

% percentage, significant of P-value < 0.5

The patency of dentinal tubules showed the ratio of scores (1,2) of it (minimal and moderate numbers of patent of dentinal tubules sequentially) was higher than other scores (0,3) sequentially in all groups, while the lowest ratio appeared in score 3 (all tubules patents) (5% in G1, 0% in both G2 and G3), Table 1.

The surface irregularities showed the ratio of score 1 (partially irregular surface) was higher than other scores (0,2,3) sequentially in all groups, where score 1 was 80% in G3, 70% in G2, 60% in G1. In contrast, the lowest ratio appeared in scores 2 (complete roughened surface) (5% in G1 and G2, 0% in G3) and 3 (roughened surface with globular projections) (5% in G1, 0% in G2 and G3), Table 1.

The intertubular micro-porosities showed the highest ratio appeared in score 1 (minimal micro-porosities) than other scores (2,0,3) sequentially in all groups, where the ratio was 65% in G3, 50% in G2, 40% in G1. In contrast, the lowest scores ratio was scored (0:absence of intertubular micro-porosities, 3:abundant micro-porosities), where the ratio of score 0 was 20% in G1, 5% in G2, 0% in G3 and the ratio of score 3 was 5% in G2, 0% in both G1 and G3 (Table 1).

The exposed dentinal tubules showed the highest ratio of scores (2: moderate number of exposed tubules, 3: all the dentinal tubules were exposed, 1: minimal number of exposed tubules) in all groups, where the ratio of score 2 (50% in G2, 35% in G3, 20% in G1) and the ratio of score 3 (65% in G1, 25% in G2, 15% in G3) and the ratio of score 1 was 45% in G3, 20% in G2, 15% in G1, while the lowest ratio appeared in scores 0 (absence of dentinal tubules) that was 5% in G2 and G3, 0% in G1 (Table 1).

Statistically, there was non-significant appearance of remaining residual dentin features under magnification 4,000x of SEM (with p-value and Fisher exact sequentially = 0.989 and 2.462 for smear layer, 0.963 and 2.805 for patency of dentinal tubules, 0.757 and 4.075 for surface irregularities, 0.217 and 7.308 for intertubular micro porosities) among three groups except exposed dentinal tubules that showed statistically significant difference with p-value = 0.012 and Fisher exact = 14.353 among three groups, Table 1.

Clinical features of dentin with 8,000x magnification in all groups

Our results showed the ratio of score 1 of the smear layer was higher than other scores (0,2,3) in all groups, where score 1 was 55% in G1, 50% in G3, 45% in G2 and the same precisely in magnification 4,000x. In contrast, the lowest score of the smear layer was score 3, the same as magnification 4,000x (5% in G2 and G3, 0% in G1), Fig. 5, Table 2.

Table 2 Comparison of clinical features of residual dentin at 8,000x magnification among groups using Fisher’s exact test.

Variance	Scores	G1	G2	G3	Fisher exact	P-value	
		N	%	N	%	N	%			
Smear layer	0	3	15	6	30	6	30	3.767	0.757
NS	
1	11	55	9	45	10	50	
2	6	30	4	20	3	15	
3	0	0	1	5	1	5	
Patency of Dentinal tubules	0	2	10	4	20	1	5	4.634	0.598
NS	
1	11	55	8	40	8	40	
2	7	35	7	35	9	45	
3	0	0	1	5	2	10	
Surface Irregularities	0	5	25	3	15	5	25	4.062	0.803
NS	
1	13	65	16	80	15	75	
2	1	5	1	5	0	0	
3	1	5	0	0	0	0	
Intertubular micro porosities	0	2	10	0	0	0	0	5.239	0.500
NS	
1	14	70	16	80	13	65	
2	4	20	3	15	6	30	
3	0	0	1	5	1	5	
Exposed dentinal tubules	0	0	0	1	5	1	5	5.191	0.522
NS	
1	4	20	3	15	8	40	
2	10	50	9	45	6	30	
3	6	30	7	35	5	25	

The patency of dentinal tubules showed the ratio of scores (1,2) of it was higher than other scores sequentially in all groups as same in magnification 4,000x, where in score one there was increased in G1 and G2, and remained constant in G3 (55% in G1, 40% in G2 and G3), in score two there was decreased in ratio in G2 and remain consistent in G1 and G3 (45% in G3, 35% in G1 and G2). The lowest ratio appeared in score 3 increased to 10% in G3 and 5% in G2, decreased to 0% G1 (Fig. 6, Table 2).

The surface irregularities showed the ratio of score 1 was higher than other scores (0,3,2) sequentially in all groups as same in comparison in magnification 4,000x, where score 1 was 80% in G2, 75% in G3, 65% in G1. In contrast, the lowest ratio appeared in score 3 (10% in G3, 5% in G2, 0% in G1) and scores 2 (5% in G1 and G2, 0% in G3) (Figs. 7, 8; Table 2).

Figure 7 SEM microphotograph of a sample under 8,000x magnification showing intertubular micro-porosities with surface irregularities.

Figure 8 SEM microphotograph of a sample under 8,000x magnification exposed dentinal tubules with a smooth surface.

The intertubular micro-porosities showed the highest ratio appeared in score one than other scores (2,3,0) sequentially in all groups as same in comparison in magnification 4,000x, where the ratio was (increased to 80% in G2 and 70% in G1, remaining constant to 65% in G3), whereas the lowest scores ratio was scores (3,0), where the ratio of score 3 was 5% in G2 and G3, 0% in G1, and ratio of score 0 was 10% in G1, 0% in G2 and G3 (Fig. 7, Table 2).

The exposed dentinal tubules showed the highest ratio of scores (2,3,1) in all groups, the same in magnification 4,000x, where the ratio of score 2 (increased to 50% in G1, decreased to 45% in G2 and 30% in G3) and the ratio of score 3 (increased to 35% in G2, reduced to 30% in G1, increased to 25% in G3) and the ratio of score 1 was decreased to 40% in G3, increased to 20% in G1, reduced to 15% in G2. In comparison, the lowest ratio appeared in scores of 0, precisely the same in the magnification 4,000x (5% in G2 and G3, 0% in G1) (Fig. 8, Table 2).

Statistically, there was a non-significant appearance of remaining residual dentin features under magnification 8,000x of SEM (with p-value and Fisher exact sequentially = 0.757 and 3.767 for smear layer, 0.598 and 4.634 for patency of dentinal tubules, 0.803 and 4.062 for surface irregularities, 0.500 and 5.239 for intertubular micro porosities, 0.522 and 5.191 exposed dentinal tubules) (Table 2).

Comparison of mean rank values of clinical features of dentin with 4,000x magnification among groups

In the present study, the value of the mean rank of the smear layer was (32.67) in G2 than others (29.95 for G1) and (28.88 for G3). Consequentially, at the same time, the mean rank of patency of tubules was 31.53 in G2, which was slightly higher than others (30.28 for G3, 29.70 for G1). Also, the micro-porosity was 32.85 for G1, higher than for others (30.45 for G3, 28.20 for G1). The mean rank value of surface irregularities was 30.80 in G3, slightly higher than others (30.48 for G2, 30.23 for G1). Meanwhile, the mean rank of exposed dentinal tubules in G1 was 39.88, higher than others (29.18 for G2, 22.45 for G3), Table 3.

Table 3 Comparison of mean rank values of clinical features of residual dentin at 4,000x magnification among groups using the Kruskal-Wallis test.

Variance	G1	G2	G3	Kruskal–Wallis	
	Min	Max	Mean rank	Min	Max	Mean rank	Min	Max	Mean rank	Chi-square	P-value	
Smear layer	0	2	29.95	0	3	32.67	0	2	28.88	0.595	0.743
NS	
Patency of Dentinal tubules	0	3	29.70	0	2	31.53	0	2	30.28	0.134	0.935
NS	
Surface Irregularities	0	3	30.23	0	2	30.48	0	1	30.80	0.017	0.992
NS	
Intertubular micro porosities	0	2	28.20	0	3	32.85	1	2	30.45	0.881	0.644
NS	
Exposed dentinal tubules	1	3	39.88	0	3	29.18	0	3	22.45	11.310	0.004
S	
Notes.

Min minimum

Max maximum

Statistically, there was a non-significant difference among groups in all four residual dentinal features with p-value, Chi square = (0.743, 0.595) for smear layer, (0.935, 0.134) for patency of tubules, (0.992, 0.017) for surface irregularities, (0.644, 0.881) for micro-porosity consequentially except the exposed tubules that appeared significant appearance among three groups (p-value = 0.004, Chi square = 11.310), Table 3.

All clinical features of residual dentin showed non-significant differences between each two groups except exposed dentinal tables that showed a significant difference between G1 (chemomechanical) and G2 (smart bur) with p-value = 0.041 and a significant difference between G1 and G3 (air-abrasion). In comparison, there was a non-significant difference between G2 and G3 with p-value = 0.198, Table 4.

Table 4 Multiple pairwise comparisons of exposed dentinal tubules at 4,000x magnification among groups using Dunn-Bonferroni.

Sample 1-Sample 2	Wilcoxon statistics	P-value	
G1-G2	2.047	0.041 S	
G1-G3	3.334	0.001 S	
G2-G3	1.287	0.198 NS	

Comparison of mean rank values of clinical features of dentin with 8,000x magnification among groups

In the present study, the value of the mean rank of the smear layer was (33.58) in G1, more consequentially than others (29.50 for G2 and 28.43 for G3). In contrast, the mean rank of patency of tubules was 35.20 in G3, which was higher than others (28.25 for G2, 28.05 for G1). Also, the micro-porosity was 34.17 for G3, more elevated than others (29.98 for G2, 27.35 for G1). While the mean rank value of surface irregularities was 32.38 in G2, which was higher than others (30.75 for G1, 28.38 for G3), at the same time, the mean rank of exposed dentinal tubules in G2 was 33.10, which was higher than others (32.45 for G1, 25.95 for G3), Table 5.

Table 5 Comparison of mean rank values of clinical features of residual dentin at 8,000x magnification among groups using the Kruskal-Wallis test.

Variance	G1	G2	G3	Kruskal–Wallis	
	Min	Max	Mean rank	Min	Max	Mean rank	Min	Max	Mean rank	Chi-square	P-value	
Smear layer	0	2	33.58	0	3	29.50	0	3	28.43	1.140	0.566 NS	
Patency of Dentinal tubules	0	2	28.05	0	3	28.25	0	3	35.20	2.555	0.279 NS	
Surface Irregularities	0	3	30.75	0	2	32.38	0	1	28.38	0.891	0.640 NS	
Intertubular micro porosities	0	2	27.35	1	3	29.98	1	3	34.17	2.500	0.287 NS	
Exposed dentinal tubules	1	3	32.45	0	3	33.10	0	3	25.95	2.316	0.314 NS	

Statistically, there was a non-significant difference among groups in all four residual dentinal features with p-value, Chi square = (0.566, 1.140) for smear layer, (0.279, 2.555) for patency of tubules, (0.640, 0.891) for surface irregularities, (0.287, 2.500) for micro-porosity, (0.314, 2.316) for exposed tubules consequentially, Table 5.

Discussion

The primary goal of the study is to use techniques alternative to traditional methods for treating active caries because we are dealing with children who could be anxious or resistant and refuse the use of local anaesthesia and caries removal with traditional methods is painful, so the pain was of primary concern. Therefore, to overcome the limitations of conventional restorative treatment, we use approaches which involve the removal of decayed tissue usually without the use of anaesthesia, and lower dental anxiety in children (more ‘patient-friendly’).

In histology research, the SEM device may take thousands of megapixel photographs of the tooth structure under study to provide good information about tooth structure and morphology (Shehadat et al., 2018). This research has resulted in the development of innovative techniques for preparing cavities to accommodate improved adhesive restoratives. These techniques are very important during cavity preparation to conserve the structure of the tooth, in addition the decreasing the pain, and cost-effect (Sharma et al., 2023).

Hegde et al. (2016) found that anaesthesia and aversion to rotational instrument noise cause psychological damage. These factors prevent children from receiving dental care, causing caries emergencies. Unfortunately, these scenarios make caries care more challenging, requiring anaesthesia (Hegde et al., 2016).

The importance of keeping healthy oral tissues and a patient-friendly approach is also becoming apparent. When feasible, preserve sound dentin and minimize invasive management (Bjørndal et al., 2019). Maintaining dentin substrate is crucial for tooth vitality following caries treatment (Alleman & Magne, 2012; Thazhatheethil et al., 2021).

In contemporary dentistry, “prevention of extension” has largely supplanted the older “extension for prevention” method. With this change, we want to keep as much of the tooth’s structure as possible and reduce damage as much as possible. Within the context of principles related to minimally invasive dentistry, this study found that chemomechanical, smart mechanical bur and air-abrasion repair offer many benefits. Various cavity preparation techniques have been available recently (Dhanvijay et al., 2023; Muliyar et al., 2014).

In this study, using different magnifications (x4,000 and x8,000) allows thorough visibility and precise measurements of the distinct features of dentin at various levels with details. Also, the previous studies used different magnifications (Kotb et al., 2016; Thakur et al., 2017; Thazhatheethil et al., 2021) to compare between different magnifications. In addition, to ensure the quality and consistency of data which may be crucial for researchers in the next studies or diagnostic purposes in caries removal and remaining dentin in dentistry.

The smear layer is not a stable structure, so it needs to be eliminated for the restorative materials and tooth structures to connect optimally, chemically and mechanically. The smear layer is demineralized to some extent by the weak acidic monomer, which is then integrated into the hybrid layer. As a result, the effectiveness of self-etching adhesive bonding is highly dependent on smear layer properties (Saikaew et al., 2022; Mohammed & Salih, 2011). In our result, in all groups with both magnifications (4,000x, 8,000x), only about 15–30% without smear layer (score 0) and for a dense layer of smear layer with visible tubules (score 2), 45–55% with a moderate layer of smear layer whereas the about 15–30% (score 1), and 0–5% for score three the dense smear layer without visible tubules. So, all methods do not entirely eradicate the smear layers. Still, the worse ratio of thick or dense smear layer was 12 of the total, which can affect the adhesion of restoration with surface dentin. This result was agreed with Thazhatheethil et al. (2021), who used two materials (Carie-careTM; Brix-3000) in the removal of caries and showed that score 1 was about 46–50% than other scores. Thakur et al. (2017) used Papacarie Duo and Carie Care for caries removal and found that the ratio of the remaining smear layer was minimal in magnification (5,000x, 1,000x). In Duruk, Kizilci & Malkoç (2022) showed a remaining thin smear layer using Carisolv and air-abrasion Kotb et al. (2016) showed that the Papacarie produced dentin surfaces with partial or complete smear layer removal, while the conventional drilling method with round carbide bur created a continuous smear layer (Kotb et al., 2016).

Statistically, there was a non-significant difference among the three groups in the ability to remove the smear layer, which agreed with Thazhatheethil et al. (2021). Somani et al. (2015) found that the smear layer removal was significantly higher in the sodium hypochlorite-based chemomechanical caries removal agents (Carisolv) than in the enzyme-based chemomechanical caries removal agent (Carie Care) and was least in the control group treated with saline. In Dhanvijay et al. (2023) showed that the Er: YAG laser showed significantly greater removal of the smear layer than papain-based chemomechanical caries removal agent on the excavated caries surfaces (p < 0.001).

In the present study, in comparison to the other results, the mean rank of smear layer presence was 29.95 (4,000x) and 33.48 (8,000x) in G1 (chemomechanical caries removal) that was near the Thazhatheethil et al. (2021) that showed the mean rank was 29.10 (5,000x)–29.12 (1,000x) in (chemomechanical caries removal with Carie-careTM) and 31.88 (5,000x)–31.90 (1,000x) in (chemomechanical caries removal with Brix-3000). In comparison, Somani et al. (2015) found that the mean rank of the presence of the smear layer was higher in the saline group and Carie Care (23) than in the Carisolv group (eight).

Because opening dentin tubules following caries removal can improve restoration bonding adhesion and lessen post-operative dentin sensitivity, the operator must choose an appropriate caries removal method and treatment option to alleviate post-operative dentin sensitivity pain and maximize restoration retention (Liu et al., 2020; Perdigão, 2020). In our study, the ratio of dentinal tubules patent was higher in the scores 1,2 (minimal, moderate patent tubules) together, ranging between 75–85% in both 4,000x and 8,000x magnification under SEM in all groups. This ratio reflects the effect of different methods in opening the dentinal tubules of teeth. This result agreed with Thazhatheethil et al. (2021), which found 80–90% of patent dentinal tubules in the chemomechanical caries removal method. Duruk, Kizilci & Malkoç (2022) showed that after caries removal, there were almost all open dentinal tubules with Carisolv, while there were partially open dentinal tubules with the Air-abrasion method. In contrast, Thakur et al. (2017) found that the using of 2 materials (Papacarie Duo; Carie Care) for caries removal appeared patent dentinal tubules in magnification (5,000x, 1,000x).

The patency of dentinal tubules differences among the three groups were non-significant in both magnifications. This result was agreed with Thazhatheethil et al. (2021). The mean rank of patency of dentinal tubules in G1 (CMCR) was 29.70 (4,000x) to 28.05 (8,000x) in our study, which was near the Thazhatheethil et al. (2021) that showed the mean rank was 29.62 (5,000x)-29.90 (1,000x) in (chemomechanical caries removal with Carie-careTM) and 31.38 (5,000x)-31.10 (1,000x) in (chemomechanical caries removal with Brix-3000).

For several reasons, it is common for bonds to be more assertive when formed on a rough surface. A larger surface area is available for the adhesive to contact while creating a connection when the surface is rough. Roughness at the interface adds to the mechanical locking. Surface irregularities may slow the spread of cracks, allowing for more robust, fatigue-resistant bond repair to experience volumetric changes of varying sizes, increasing the microporosity of the intertubular dentin (Koodaryan, Hafezeqoran & Poursoltan, 2016; Varadan, Balaji & Kandaswamy, 2015). In our result, the ratio of partial surface irregularities of dentin was non-significant higher in G3 (air-abrasion) than G2 (smart bur) and G1 (chemomechanical) in both magnifications where the ratio of score 0 (smooth surface) was 25–30% (G1), 15–25% in both G2 and G3, while the ratio of score 1 (partially irregular) was 25–30% in G1, 60–65% in both G2 and G3. These scores (0,1) were higher than other scores in all groups. This result was in agreement with Thazhatheethil et al. (2021), where the ratio of scores (1) was 56.7% (5,000x)-66.7% (1,000x) in Carie Care group and 66.7% (5,000x)-80% (1,000x) in Brix-3000 group that were higher than other scores.

The mean rank of surface irregularities presence was 30.23 (4,000x) and 30.75 (8,000x) that was near the Thazhatheethil et al. (2021) showed the mean rank was 27.73 (5,000x)-27.43 (1,000x) in (Carie-careTM) and 33.27 (5,000x)-33.57 (1,000x) in (Brix-3000).

The higher proportion of intertubular microporosities adversely affects the adhesive bonding effectiveness (Isolan et al., 2018). Our result showed that minimal micro-porosity (score 1) than other scores in all three groups, where the ratio was 40–65% in 4,000x, 65–80% in 8,000x, while the score 2 (moderate porosity) was 35–40% in 4,000x, 15–30% in 8,000x. This result was in agreement with Thazhatheethil et al. (2021), where the ratio of scores (1) was 40% (5,000x)–46.7% (1,000x) in Carie-careTM group and 56.7% (5,000x)–60% (1,000x) in Brix-3000 group that were higher than other scores. These differences were non-significant among the three groups in both studies.

The mean rank of the presence of intertubular micro porosities in the chemomechanical group was 28.20 (4,000x) and 27.35 (8,000x), which was non-significantly lesser than other groups (smart bur and air-abrasion). At the same time, this result was near to the findings of Thazhatheethil et al. (2021) that showed the mean rank was 31.10 (5,000x)–30.18 (1,000x) in (Carie-careTM) and 29.90 (5,000x)–30.82 (1,000x) in (Brix-3000).

Exposed dentinal tubules facilitate robust micro-retention, making them suitable for adhesive restorations (Dhanvijay et al., 2023). According to the present study, the ratio of exposed dentinal tubules in G1 was higher than in other groups, where with magnification 4,000x, the ratio of score 3 (all dentinal tubules exposed) 65% in the chemomechanical group, 50% in score 2 (moderate number of exposed tubules) in smart bur group, and 45% in score 1 (minimal number of exposed tubules) in an air-abrasion group. Whilst with 8,000x, the ratio of score 2 was higher in groups (chemomechanical and smart bur) (50% and 45% respectively) and 40% in score 1 in an air-abrasion group. This result was agreed with Thazhatheethil et al. (2021) that showed that the score 3 was 50% (5,000x)-60% (1,000x) in the Carie Care group, score 2 was 43% (5,000x)- 40% (1,000x) in the Brix-3000 group.

The exposed dentinal tubules showed a significant difference among the three groups in 4,000x magnifications (p-value = 0.012) and non-significant in 8,000x magnifications (p-value = 0.522). At the same time, in 4,000x magnification, the chemomechanical group showed significant differences with other groups (smart bur, air-abrasion) with p-value = 0.041 and 0.001, respectively. The smart bur and air-abrasion relationship was non-significant (p-value 0.198). This result disagreed with Thazhatheethil et al. (2021), which showed a non-significant difference between the two groups (Carie Care; Brix-3000). This difference was related to the group’s selection in both studies, where our study included three different methods with different mechanisms of working in caries removal. In contrast, Thazhatheethil et al. (2021) had only a single method of caries removal that was chemomechanical using two different materials. Duruk, Kizilci & Malkoç (2022) showed more exposed dentinal tubules in the Carisolv gel group than in the laser group.

In 4,000x magnification, the mean rank of exposed tubules in the chemomechanical group was 39.88 and 33.10, with 8,000x in the smart bur group, which was non-significantly higher than other groups. Thazhatheethil et al. (2021) showed 33.28 in Carie-careTM, 27.72 in Brix-3000 with 4,000x, whereas the 8,000x showed 33.75 in Carie-careTM, 27.25 in Brix-3000.

A 2023 study by Sharma et al. (2023) found no statistically significant difference in the efficacy of the various techniques in eradicating caries, although bur excavation was faster than chemomechanical.

Research has shown that caries removal agents work similarly to Ca(OH)2 pulp capping materials, and using chemical agent gel had no adverse effects on dental pulp tissue. The alkalinity of the caries removal agent was also linked to its hemostatic and antibacterial effects on tooth pulp, which were also discovered (Hamama, Yiu & Burrow, 2014; Aubeux et al., 2021). This may show that these medicines effectively eliminated the diseased carious dental tissue (Puri et al., 2020).

When working with enamel, it is necessary to use a combination of polymer burs and steel or diamond burs because the technique is sensitive, high-pressure application excessively excises healthy dentin, and the burs break when they come into direct contact with the enamel. However, mechanical excavation using rotary instruments is not selective when removing caries; it also causes pain, increases vibrations and contact pressure, puts the surrounding tissues at risk of overheating, and makes a stressful, unpleasant noise. Accessing the carious tissues may be the sole goal of traditional rotary devices in minimal intervention dentistry (MID) procedures (Frencken et al., 2012). During caries excavation, the loss of tactile sensation is the fundamental disadvantage of air abrasion (Tan, Hill & Anderson, 2015).

The result of this study supports our hypothesis and suggests that samples prepared using three methods (chemomechanical, mechanical, and air abrasion) exhibited no significant differences in surface morphology of residual dentin remained after caries removal in 4,000x and 8,000x magnifications except dentin exposed that showed significant higher ratio in a chemomechanical group than other groups in magnification 4,000x. The study had some limitations, including that it only examined primary molars and did not evaluate the residual bacterial deposits left behind following caries eradication.

However, in this study, no feedbacks were taken from the patients about the comfort of the technique used with them and this point can be taken in future papers.

Conclusions

The minimal intervention (CMCR, MCR, and AACR) approaches can be utilized in restorative dentistry by preserving healthy dental tissues and extending the life of the natural teeth to their fullest potential. These approaches are deemed effective methods of caries elimination based on the evidence (smear layer, tubules openness, surface irregulates and others). Caries removal takes time, but it helps patients who are extremely anxious or frightened.

Supplemental Information

Supplemental Information 1 Raw Data 4,000x

Supplemental Information 2 Raw Data 8,000x

The authors would like to thank Mustansiriyah University, College of Dentistry, Baghdad–Iraq for their moral support.

Additional Information and Declarations

Competing Interests

Author Contributions

Human Ethics

Data Availability

The authors declare there are no competing interests.

Maha Abdul-Kareem Mahmood conceived and designed the experiments, performed the experiments, analyzed the data, authored or reviewed drafts of the article, and approved the final draft.

Haraa Khairi Al-Hadithi performed the experiments, authored or reviewed drafts of the article, and approved the final draft.

Hashim Mueen Hussein conceived and designed the experiments, analyzed the data, prepared figures and/or tables, authored or reviewed drafts of the article, and approved the final draft.

The following information was supplied relating to ethical approvals (i.e., approving body and any reference numbers):

The study was done after approval from the Committee of the College of Dentistry in the University REC reference REC135, study number MUPEDO3

The following information was supplied regarding data availability:

The raw data are available in the Supplemental Files.

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
