# Peer review of "Evaluation of dentin features in teeth after caries removal by three techniques (chemomechanical, mechanical with a smart bur, and air-abrasion): an in vitro study"

_PeerJ, doi:10.7717/peerj.17717_

## Round 0.1 · original submission · Major Revisions

I would like you to pay particular attention to the concerns over experimental design that Reviewer 1 identified related to traditional methods of caries removal. I understand that these alterations to the data might not be possible at this stage, but it is essential that you, at minimum, express why you exclude traditional caries removal methods from this study.

Other comments from the reviewers are important as well, though arguably not as major.

**Language Note:** The review process has identified that the English language must be improved. PeerJ can provide language editing services - please contact us at [email protected] for pricing (be sure to provide your manuscript number and title). Alternatively, you should make your own arrangements to improve the language quality and provide details in your response letter. – PeerJ Staff

Reviewer 1 ·

Basic reporting

The manuscript lays out a solid comparison of three ways to get caries out, shining a light on how they mess with dentin. Using scanning electron microscopy to get into the nitty-gritty, like the smear layer, tubule openness, and how bumpy the surface is, really adds depth to the analysis. But, going deeper into the stats could make your conclusions firmer. Also, making the SEM pics clearer and giving more reason behind picking those caries removal methods could polish up the paper. Tying up with direct feedback on patient comfort or their prefs could lend more practicality to your findings. Tackling these bits would majorly boost what your manuscript brings to the table, turning it into a richer resource for folks keen on less invasive dentistry.

Experimental design

On page 9, where Fisher's exact test and the Kruskal-Wallis tests are mentioned, it'd be really useful to dive deeper into why these tests were specifically picked, given the way your data's spread out and what hypotheses you were poking at. This'll let readers get why you went with these stats methods for your data.

Discussing the significance levels (p-value less than 0.05) could really do with some more meat on the bones, especially talking about the study's power and the sizes of effects noticed. This becomes super critical where you've got differences that don't pop out as significant, to figure if it's down to not having enough data points or if it's genuinely no biggie (see page 10).

Those SEM images you've got are key for getting the lowdown on the dentin stuff after taking the caries out. But, you could make the paper a bit clearer by throwing in a scale bar on all SEM snaps (Figures 4, 5, 6, 8) to give readers a solid grasp on the size of what they're looking at.

The paper could do with a bit more on why you chose those three specific techniques for yanking caries. Fleshing this out in the intro or methods section would make the logic flow better and give folks a clearer picture of your study's design and why it matters for clinic work.

Validity of the findings

Wrapping up on how effective and comfy the caries removal methods are would hit harder if you had feedback from patients or some numbers on how comfy they were. If you didn't pick up this kind of data, maybe talk about it being a gap and something to chase up in future studies.

Reviewer 2 ·

Basic reporting

The title may be altered to “Evaluation of dentin features in teeth after caries removal by three techniques : an In vitro study”.
The English writing ought to be improved to ensure that an international audience can clearly understand text. For instance, in line 36 the objectives in the abstract should use “this study” rather than “this paper”. From line 73 to 74, the meaning of these two sentences should be elucidated.
From line 59 to 60, the description of "can successfully treat patients of
60 young ages, as well as scared and stressed patients." was not appropriate, for it was not a clinical trial.
In line 83, since it is the first appearance of “CMCR”, the abbreviation should be elucidated.
The language deserves major revision to conform to a more formal scientific writing style. 
The description of “a cross-sectional study” is not appropriate, because it was an in vitro study.
Figure 3 should be modified. A schematic diagram may be preferred to illustrate the mechanism of this method.

Experimental design

A commonly used standard method for caries removal, such as traditional metal bur should be included in the design to confirm the relative efficacy of the new three methods.

Validity of the findings

Though the results showed all groups were effective in removing caries, the efficacy could not be determined without a comparison to traditional methods.

Additional comments

More details of the new technologies should be introduced in the background of the abstract.
The results presentation seemed unclear, the authors should concentrate on the main points. For example, from line 215 to 245, the authors spent a large portion describing the scores in every group instead of focusing on the statistical results among the three groups.
The discussion was tediously long which should be modified to be more concise.

---

## Round 0.2 · Minor Revisions

Dear authors,

After taking the reviewers' comments into consideration I suggest publication of your manuscript with the condition of making the following changes:

1. Please clarify in the title and in the manuscript that your study is comparing three techniques for dentin removal. This is to lift the concern that you have not used the "golden standard" (a slow speed metal bur) in your investigation.

2. Justify the use of two different magnifications, namely x4000 and x8000. In my view there is no need to use both and I would probably keep one in order to avoid potential confusion by the readership regarding the interpretation of the results. My suggestion is to either report in the main manuscript the results of one magnification (and report the other results as a supplement) OR justify the reporting of both magnifications in the main manuscript.

Kind regards.

Reviewer 1 ·

Basic reporting

After reviewing the revised manuscript, I'm convinced the changes have significantly improved its quality, meeting the publication standards. The effort to address previous concerns is commendable, and the manuscript now presents a solid case for acceptance.
I fully support its publication and look forward to its contribution to our field.

Experimental design

none

Validity of the findings

none

Reviewer 2 ·

Basic reporting

no comment

Experimental design

Still, the author was not able to explain clearly the rationale for not using traditional methods as comparison.

Validity of the findings

Without comparison to the traditional methods, the conclusion should be altered to be humbler.

---

## Round 0.3 · accepted · Accept

Thank you for addressing the concerns of the reviewers. I have now suggested publication of the work.